# Molecular Docking and Intracellular Translocation of Extracellular Vesicles for Efficient Drug Delivery

**DOI:** 10.3390/ijms232112971

**Published:** 2022-10-26

**Authors:** Yasunari Matsuzaka, Ryu Yashiro

**Affiliations:** 1Division of Molecular and Medical Genetics, Center for Gene and Cell Therapy, The Institute of Medical Science, The University of Tokyo, Minato-ku 108-8639, Tokyo, Japan; 2Administrative Section of Radiation Protection, National Institute of Neuroscience, National Center of Neurology and Psychiatry, Kodaira 187-8551, Tokyo, Japan; 3Department of Infectious Diseases, Kyorin University School of Medicine, 6-20-2 Shinkawa, Mitaka-shi 181-8611, Tokyo, Japan

**Keywords:** cancer, exosomes, microRNAs, immune system, molecular docking, drug delivery

## Abstract

Extracellular vesicles (EVs), including exosomes, mediate intercellular communication by delivering their contents, such as nucleic acids, proteins, and lipids, to distant target cells. EVs play a role in the progression of several diseases. In particular, programmed death-ligand 1 (PD-L1) levels in exosomes are associated with cancer progression. Furthermore, exosomes are being used for new drug-delivery systems by modifying their membrane peptides to promote their intracellular transduction via micropinocytosis. In this review, we aim to show that an efficient drug-delivery system and a useful therapeutic strategy can be established by controlling the molecular docking and intracellular translocation of exosomes. We summarise the mechanisms of molecular docking of exosomes, the biological effects of exosomes transmitted into target cells, and the current state of exosomes as drug delivery systems.

## 1. Introduction

Intracellular communication mediated via extracellular vesicles (EVs) and drug-delivery systems based on EV biology have emerged as areas of investigation with a significant potential to impact human health [1,2,3,4]. EVs are comprised of lipid bilayer membranes and are classified into exosomes, microvesicles, and apoptotic bodies, mainly based on differences in their biogenesis [5,6,7,8]. Microvesicles emerge from the plasma membrane of the cell and have diameters of approximately 100 nm to 1000 nm [9,10]. Apoptotic bodies are giant vesicles formed from the plasma membrane during the induction of apoptosis and are approximately 1–5 μm in diameter [11,12,13]. Exosomes are formed within the intracellular multivesicular endosome (MVE) and are approximately 30–150 nm in diameter [14,15,16,17,18,19,20,21,22,23,24,25,26,27,28,29,30]. EVs are secreted by almost all cells. The release of the exosomes into the extracellular space is mediated by the fusion of the MVE membrane with the plasma membrane. Exosomes are found in various body fluids, such as plasma, serum, urine, saliva, and breast milk, and are also released in the culture supernatants of many cell lines [31,32,33,34]. They contain functional molecules, such as nucleic acids, microRNAs (miRNAs), messenger RNA (mRNA), and circulating RNA (circRNAs); metabolites and lipids on their membrane, such as phosphatidylserine; proteins, such as enzymes; and are characterized by some antigens, such as CD9, CD63, and CD81 [35,36,37,38,39,40,41,42,43,44,45,46,47]. These molecules are important for the characterisation of the cell type of origin for a given population of exosomes [48,49,50,51,52,53,54,55,56,57,58,59,60,61,62,63,64,65,66,67,68,69,70,71,72,73,74,75,76,77,78,79,80,81,82,83,84,85,86,87,88,89,90,91,92,93,94,95]. Secreted vesicles are transported by body fluids, such as blood, to cells where their impact is likely to be the greatest. The release of vesicles mediates intercellular communication [48,49,50,51,52,53,54,55,56,57,58,59,60,61,62,63,64,65,66,67,68,69,70,71,72,73,74,75,76,77,78,79,80,81,82,83,84]. In summary, exosomes reflect the characteristics of the cell from which they were derived, are taken up by other cells, and are responsible for the intercellular transmission of information. In this review, we aim to summarise the intercellular mechanisms of exosomes, the biological effects of exosomes transmitted into target cells, and the current state of exosomes as drug-delivery systems. In addition, we summarize the latest studies on EVs as novel drug-delivery systems and their effects on cancer.

## 2. EVs in Cancer Metastasis and Malignant Transformation

EV-mediated intracellular communication plays a role not only in maintaining cell homeostasis, but also in disease progression [96,97,98,99]. Exosomes are associated with several diseases, including cancer metastasis, which is the invasion and spread of primary cancer cells into other organs [100,101,102,103,104,105,106,107,108,109,110,111,112,113,114,115,116,117,118,119,120,121,122,123,124,125,126,127,128,129,130,131]. Exosomes have been shown to play an important role in the formation of the cancer microenvironment and the mechanisms of cancer malignancy, such as cancer cell growth, infiltration, metastasis, and pre-metastasis niche [132,133,134,135,136,137,138,139,140]. Tumour-derived miRNA exosomes have been shown to affect tumour cells and stromal cells of the tumour microenvironment, such as fibroblasts and macrophages [141,142,143,144,145]. For example, miRNA exosomes secreted by cancer cells induce intratumoural angiogenesis, thereby promoting metastasis [145,146,147,148,149]. Moreover, miR-181c promotes the delocalization of actin fibres by downregulating the 3-phosphoinositide-dependent protein kinase 1 (PDPK1) gene [150].

Cancer-derived exosomes disrupt the blood–brain barrier and promote brain metastasis [151,152,153,154]. In breast cancer, tumour-derived exosomes containing miR-181c have been shown to promote cancer metastasis to the brain by disrupting the blood–brain barrier [150]. In addition, exosomes secreted by ovarian cancer containing matrix metallopeptidase 1 (MMP1) mRNAs induce peritoneal mesothelial cells to undergo apoptotic cell death, thereby promoting peritoneal cancer metastasis [155].

## 3. EV-Mediated Immune Escape of Cancer Cells

Tumours evade the immune systems by secreting exosomes containing proteins that suppress the immune response. A new pathway for such immune evasion has been identified in a laboratory model of skin cancer melanoma and patients suffering from the disease. Tumour cells release exosomes coated with programmed death-ligand 1 (PD-L1) proteins, which are immune checkpoint proteins that bind to immune cells to inactivate them [156,157,158,159,160,161,162,163]. This prevents immune cells from reaching tumour cells and attacking them. [164,165,166,167,168,169,170,171,172,173,174,175,176,177,178]. PD-L1 is present in exosomes released from melanoma cells [178,179,180,181,182,183,184,185,186]; however, exosomes derived from metastasised melanoma cells have a higher PD-L1 content than those derived from primary focal melanoma cells. Moreover, an electron microscope analysis revealed the PD-L1 protein was carried on the surface of the protein [187]. This suggests that PD-L1 on the surface of exosomes interacts directly with the immune cells. Furthermore, PDL1-positive exosomes mainly bind to cytotoxic T cells, preventing their proliferation and attack on cancer cells [159,188]. In a mouse model of melanoma that closely mimics human cancer, the injection of PDL-1-coated exosomes promoted tumour growth and reduced the number of T cells and other immune cells in and around the tumour (Figure 1) [162,173,189,190,191,192]. In addition, exosomes carrying PD-L1 have been identified in blood samples of patients with a history of breast cancer, melanoma, or lung cancer who had received treatments for their respective diseases [193,194,195].

Melanoma elicits a particularly strong immune response, and multiple immune checkpoint inhibitors have been approved by the US Food and Drug Administration for the treatment of melanoma [163,164,165,166,167,168,169,170]. Interestingly, based on PD-L1 levels in exosomes, patients who are most likely to respond to checkpoint inhibitors can be identified and the response to these drugs can be evaluated [162,194,196,197,198,199]. For example, compared to patients with high exosomal PD-L1 levels before treatment, those with lower levels responded remarkably better to treatments with the checkpoint inhibitor pembrolizumab (Keytruda), which can block PD-1, the immune cell binding partner of PD-L1 [185,196,198,200,201,202,203,204,205]. In contrast, after the start of treatment, higher exosomal PD-L1 levels reflected a reduction in tumour size [206,207,208], indicating that two different mechanisms occur. Before treatment, exosomal PD-L1 levels likely reflect the size of the tumour and the extent of the disease. In other words, a high blood PD-L1 level indicates the presence of several tumours and is associated with a poor prognosis. After treatment, a rapid increase in exosomal PD-L1 in patients who responded to treatment indicates that T cells are activated and secrete more cytokines, such as interferon-γ (IFN-γ), which are signalling molecules that can stimulate the immune system [162,191,209,210,211]. In melanoma cell lines, treatment with IFN-γ-has been shown to increase exosomal PD-L1 [162,212]. Moreover, analysis of patient samples revealed that exosomal PD-L1 levels tended to increase or decrease with increasing IFN-γ levels. Therefore, these findings also indicate that levels of exosomal PD-L1 in blood may help in selecting the appropriate treatment for different individuals. However, the exact mechanism by which exosomes carrying PD-L1 affect the immune response to tumours in patients with melanoma remains uncertain. Moreover, other types of immunomodulatory molecules may be present on the surface of these exosomes. Therefore, further research is needed using more samples from patients with melanoma and other cancer types and close comparison of PD-L1 in tumour biopsies with exosomes released from tumours should be observed. For example, approximately 40% of human melanoma cells express significant amounts of PD-L1 on their surfaces [213]. The presence of large amounts of exosomal PD-L1 in the blood of melanoma patients also suggests that PDL-1 has an overall effect on immunosuppression in those patients [161,162,182,186,214]. However, presently, there is no evidence that the immunity of patients with stage 4 melanoma is impaired.

For many years, exosomes have been thought to only function as molecular carriers that carry waste from cells; however, it has become clear that EVs affect various biological processes and diseases, including immune responses and cancer [215,216,217,218,219,220,221,222,223,224,225,226,227,228,229,230,231,232]. Nonetheless, it is difficult to identify these exosomes. Moreover, the capacity of each exosome is only one-millionth of that of a typical cell, and most modern biomedical research tools are not suitable for the accurate and functional analysis of cargos within individual exosomes. Therefore, new tools and approaches would be useful to study these small vesicles in more detail.

## 4. Intracellular Translocation of Nucleic Acids and Proteins via EVs

EVs are highly expected to be next-generation drug carriers for the following reasons: (1) possible immunoregulation, (2) possible expression of membrane proteins by genetic engineering, (3) intracellular communication pathways, (4) low cytotoxicity, and (5) infinite secretion [1,233,234,235,236,237,238,239,240,241]. For example, small interfering RNA (siRNA) delivery for a BACE (β-site of Amyloid Precursor Protein cleaving enzyme) target using exosomes has proven to be beneficial in the treatment of Alzheimer’s disease [242,243,244]. Furthermore, because exosomes that have miRNAs and enzyme-encapsulating cocktails, which have functions such as cell proliferation suppression and cell migration promotion, are naturally secreted from cells, they are highly expected to be used as drugs for treating several diseases [245,246,247,248,249,250,251,252]. The macropinocytosis pathway is important for the intracellular translocation of exosomes [253,254,255,256,257,258]. Therefore, by modifying the membrane surface of exosomes with a functional peptide to induce macropinocytosis, it is possible to considerably increase the efficiency of the intracellular delivery of exosomes [254,255,257].

## 5. EV Uptake in Target Cells via Macropinocytosis

Eukaryotic cells take up molecules, such as extracellular proteins and lipoproteins, by a process called endocytosis, which includes clathrin-dependent or -independent endocytosis and micropinocytosis (Figure 2) [259,260,261,262,263,264,265,266,267,268,269]. In clathrin-dependent endocytosis, when a ligand molecule binds to a receptor on the plasma membrane, a clathrin molecule binds to the cytosol via an AP2 adaptor protein, and a ball-shaped structure of the plasma membrane is formed [270,271,272,273].

In addition, dynamin then separates the endosome from the plasma membrane via clathrin-dependent endocytosis [274,275,276]. These formed endosomes usually measure up to approximately 120 nm in diameter because clathrin limits the size of endosomes [277,278,279] Therefore, in normal clathrin-dependent endocytosis, the intracellular translocation of EVs is inefficient. By contrast, macropinocytosis can take up extracellular molecules with a diameter greater than 1 mm, including nutrients, into cells. Macropinocytosis is a clathrin-independent pathway, characterised by actin-dependent reorganisations (lamellipodia) of the plasma membrane to form macropinosomes [280]. Macropinocytosis is induced by the activation of various receptors, such as the epidermal growth factor receptor (EGFR), which is highly expressed in tumour cells, such as human epidermal cancer A431 and the chemokine receptor CXCR4 [253,281,282,283,284,285,286,287,288,289,290,291,292,293,294,295,296,297].

Furthermore, in vitro experiments have revealed that human pancreatic cancer-derived MIA PaCa cells, which highly induce macropinocytosis, have high exosomal migration efficiency [253,298,299]. Normally, the exosome membrane is negatively charged (zeta potential is approximately −10 mV); therefore, it repels with the negatively charged plasma membrane [300,301,302,303]. However, in the macropinocytosis pathway, exosomes that do not easily interact with the plasma membrane can be effectively wrapped and incorporated into cells using ruffling [296,304]. These findings highly suggest that macropinocytosis contributes to intercellular communication.

In pancreatic cancer, exosomes derived from normal fibroblast-like mesenchymal cells were engineered to carry siRNA or shRNA specific to oncogenic KRAS, which is a common mutation in pancreatic cancer. These exosomes were effectively taken up by target cells by macropinocytosis and exerted a remarkable effect in suppressing pancreatic cancer cell growth in vivo [281,305,306,307,308]. Moreover, in lung cancer, gefitinib (Iressa), an anticancer drug that inhibits the epidermal growth factor receptor (EGFR), was shown to increase the intracellular translocation of exosomes, but decrease that of liposomes, which are typically used for drug-delivery systems [309]. In addition, compared to liposome-encapsulated doxorubicin, exosome-encapsulated doxorubicin results in robustly higher anticancer activity against non-small cell lung cancer [310,311]. Furthermore, degradation of the exosomal membrane proteins further enhanced intracellular migration induced by gefitinib treatment [304]; thus, exosomal membrane lipids may contribute to the promotion of intracellular translocation during gefitinib treatment. Additionally, a macropinocytosis inhibitor was shown to remarkably suppress the growth of pancreatic cancer cells both in vitro and in vivo [288,289,290,291,292]. Thus, macropinocytosis pathway inhibitors can be used to suppress the progression of pancreatic cancer. These findings suggest that drug applications from a new perspective are highly beneficial to treat cancer.

## 6. Techniques for Functional Peptide Modifications on the Exosomal Membrane

A previous study on the intracellular uptake of exosomes using a functional peptide modification technique on the exosomal membrane has revealed its importance for the intracellular translocation of exosomes [258,298,312]. Two simple techniques for binding peptides to the exosomal membrane without modifying the components of the membrane have been reported: (1) acylating a peptide such as a stearyl group and (2) using a peptide with a linker including a succinimide group [298,313]. In the method of acylating a peptide with a stearyl group, when the peptide is synthesised on beads by the Fmoc solid phase method, the N-terminal is dehydrated and condensed with stearic acid, and then deprotected and purified to obtain the acylated target peptide [254,258,314]. For example, a peptide with a stearyl group can easily be inserted into an exosomal membrane with an acylated hydrophobic part, and the peptide can be presented on the exosomal membrane simply by mixing the peptide with the exosome in a solution [258,315]. Peptides that are not present in the exosome membrane can be removed by ultrafiltration. This method does not require consideration of the sequence of the peptide; however, for highly hydrophobic peptides, the solubility may deteriorate owing to the addition of a hydrophobic group to the peptide. In such cases, the balance between the control of the hydrophobic group to be acylated and the degree of insertion into the exosome membrane should be considered. Furthermore, in the method using a peptide with a linker including a succinimide group, it is possible to covalently bind the target peptide and the exosomal membrane protein using a divalent linker, such as N-(6-maleimidocaproyloxy) sulphosuccinimide and sodium salt (sulpho-EMCS) [257]. During peptide synthesis, after introducing cysteine residues into the peptide sequence, an acetylation cap with acetic anhydride is applied at the N-terminal cysteine residue side chain of the purified target, and the maleimide group of EMCS is bound by Michael addition and purified again. Next, by mixing the purified peptide and exosome, the succinimide group possessed by the linker of the peptide reacts with the membrane protein amino group of the exosome, and the target peptide and exosome are covalently attached. This method can be used when an amino group, such as lysine or a cysteine residue, does not exist in the peptide sequence to be originally bound. However, it may be necessary to introduce an unnatural amino acid into the amino acid sequence and use a linker bond using click chemistry. When the first method is used, the membrane protein of the exosome is hardly affected and the peptide can be modified, whereas in the second method, since a covalent bond is formed on the side chain of the constituent amino acids of the membrane protein, the original function of the membrane protein may be affected. However, because the second method modifies the peptide by covalent bonding, peptide retention on the exosomal membrane is higher than that of the anchor type inserted into the membrane using the first method. Therefore, it is evident that the macropinocytosis pathway is important for the intracellular translocation of exosomes [253]. The development of a technique that can induce macropinocytosis using exosomes can enhance intracellular migration.

A membrane-permeable arginine peptide-modified exosome that induces macropinocytosis has been developed. In addition, the human immunodeficiency virus (HIV)-1 encodes the transcription factor trans-activator of transcription (Tat) protein-derived peptides and oligoarginines, which are cell-penetrating peptides that can easily penetrate cells (CPPs) [316]. CPPs contain many arginine residues in their sequence; therefore, they accumulate in the plasma membrane because of their interactions with heparan sulphate of the sugar chain of proteoglycan in the membrane. As a result, clustering of proteoglycan (syndecan-4), intracellular binding of protein kinase C, alpha (PKCα) to proteoglycan, and signal transduction occur. Moreover, EVs can be efficiently taken up into cells by the activation of the small G protein Rac1, which induces macropinocytosis due to the remodelling of the actin skeleton [316]. Therefore, exosomes induce macropinocytosis in target cells by binding to membrane-permeable arginine peptides. Moreover, the EV uptake efficiency depends on the number of peptides bound to the exosome membrane. When octaarginine (stearyl-R8), which is a typical membrane-permeable arginine peptide with the previously mentioned stearyl group at the N-terminus, is mixed with CD63-GFP (green fluorescent protein)-exosomes to modify the peptide on the exosome, exosomes can act as scaffolds to promote the clustering of syndecan-4 on the plasma membrane of target cells and induce macropinocytosis with foliate pseudopodia by modifying stearyl-R8. This can remarkably increase the efficiency of the intracellular translocation of exosomes. Notably, this method caused almost no cytotoxicity [258].

In addition, the clustering of proteoglycans by the peptide and the induction of macropinocytosis are affected by the number of arginine residues in the sequence [316,317]. Furthermore, the binding of oligoarginines with different numbers of arginine residues to the membrane surface of CD63-GFP-exosome does not affect the morphology of the exosomes. Notably, this method did not exhibit cytotoxic effects. In addition, as previously mentioned, despite the negative charge of the exosomal membrane, when oligoarginine and exosomes were simply mixed without using a divalent linker, no increase in the efficiency of intracellular translocation of exosomes was observed with any oligoarginine [257]. Thus, the strong binding of a functional peptide to the exosomal membrane using the method previously discussed is important for fully exploiting the functionality of the peptide. It was also reported that the activity of drug-encapsulated exosomes was markedly higher in exosomes bound with the R16 peptide, which has a relatively low intracellular translocation compared with those bound to R8 and R12 peptides, which have a higher translocation. This suggests that the cytosolic release efficiency of modified exosomes after intracellular translocation is high, although R16 peptides have lower intracellular translocation than the R8 and R12 peptides [257]. Therefore, when selecting a functional peptide, a well-balanced peptide must be selected by considering not only the intracellular transfer efficiency, but also the release efficiency after intracellular transfer.

Normally, when the drug is delivered intracellularly by endocytosis or macropinocytosis, the contents of the endosome are degraded by various enzymes [318,319]. Therefore, the drug must escape into the cytosol before lysosomal degradation. Nakase et al. have developed an efficient cytosolic delivery technique for proteins using the GALA peptide, which is a pH-sensitive membrane fusion peptide [320,321]. The GALA peptide (amino acid sequence: WEAALAEALAEALAEHLAEALAEALEALAA) is an artificial peptide that mimics the membrane fusion protein of a virus composed of 30-residue amino acids. When the pH is neutral, the peptide has almost no secondary structure; however, as the pH decreases, the helix content increases, facilitating the incorporation of the peptide into the membrane, which promotes membrane destabilisation and fusion [320,321]. This GALA peptide is rich in glutamate residues and negatively charged. However, since the cell plasma membrane is also negatively charged, the intracellular transferability of the GALA peptide alone is extremely low. Therefore, the formation of a complex between a cationic lipid and GALA significantly increased the intracellular translocation of the GALA peptide. Furthermore, by binding molecules, such as proteins to be carried to the cytosol, to the GALA peptide and forming a complex with the cationic lipid, this complex is taken up into cells by endocytosis, and the target molecule bound to the GALA peptide can be effectively released into the cytosol [321,322]. Although the use of a large number of cationic lipids causes cytotoxicity, cells hardly uptake GALA even if they uptake exosomes; therefore, complex formation using lipids is important. Furthermore, when ammonium chloride was used to suppress the decrease in pH in endosomes, the cytosolic escape effect of the GALA peptide was markedly reduced. This finding indicates that endosome maturation is important for peptide function. In addition, it is important to optimise the concentration in complex formation because the difference in the concentration of the GALA peptide affects complex formation and cytosol escape efficiency. This epoch-making method can easily promote the cytosolic release of exosome-encapsulated molecules by simply mixing them with exosomes. Therefore, it can be applied to various drug-encapsulating exosomes (including miRNAs) in the future, such as miRNA-encapsulating exosomes in myocytes [323].

## 7. Storages of EVs as a Long-Term Strategy

It is important to establish a preservation method for drug-encapsulating exosomes. Exosomes can be stored in a refrigerator for approximately 1–2 weeks at the most but can be frozen for several months. However, repeated freezing and thawing are known to have a serious effect on the morphology of exosomes. For preservation methods, such as freeze-drying exosomes bound to R16 peptide and then adding water to restore them, the R16 peptide-modified exosomes were shown to be hardly affected [260]. In contrast, for R16-bound exosomes containing drugs, as described above, freeze-drying resulted in a considerable reduction in drug activity. Therefore, the efficiency of the cytosolic release of the encapsulated drug after intracellular transfer may be reduced after lyophilisation. The storage capacity of EVs can depend on their number, size, function, temperature, duration, and freeze–thaw cycles [324,325,326,327,328,329].

## 8. Clinical Application

EVs are attracting attention as drug carriers that deliver nucleic acid medicine, which is a general term for medicines that do not directly encode proteins, but directly act on DNA, RNA, or chemically synthesized oligonucleotides on proteins, to the affected area [330,331,332,333,334,335]. Types of nucleic acid drugs include antisense nucleic acids, siRNA, miRNA, decoy nucleic acids, nucleic acid aptamers, etc. These are attracting attention as new therapeutic agents following low molecular weight drugs and antibody drugs that have already been clinically applied. On the other hand, nucleic acid drugs are easily degraded in blood, which contains many digestive enzymes. Therefore, in order to obtain clinical efficacy against various diseases through blood, a drug carrier that is safe to the body, prevents degradation by digestive enzymes, and selectively delivers nucleic acid medicines to the affected area is required. EVs are stable in blood due to a lipid bilayer membrane. They are about 100 nm in size, and renal excretion, immune mechanism, and enhanced permeation and retention effect are optimal. EVs are secreted by various cells in the body, and compared to artificially constructed drug carriers, immune reactions are less likely to occur. Clinical trials of EVs have already been conducted and their safety has been investigated. It is suggested that EVs are transported in an organ-specific manner. In fact, many researchers have already begun to report on in vivo studies in which nucleic acid drugs, such as siRNA, are encapsulated in EVs [336,337,338,339,340]. Furthermore, treatment methods targeting EVs are being investigated, and some are already undergoing clinical trials. The main therapeutic strategies are roughly divided into two: one is a method of targeting and removing/inhibiting harmful EVs themselves, and the other is a method of utilizing beneficial EVs. The former is thought to inhibit EVs secretion, remove EVs present in the blood, and inhibit EVs uptake. Further, Nishida-Aoki et al. reported that the EVs removal by tail vein administration of antibodies to xenograft mouse models using human cancer cell lines can suppress cancer metastasis [341]. These results suggest that EVs-targeted therapy may have clinical efficacy. The latter mainly includes the use of EVs loaded with antigenic peptides as vaccines and the use of mesenchymal stem cell (MSC)-derived EVs. In particular, MSC-derived EVs have already been clinically tested for steroid-refractory graft-versus-host disease, and there have been reports of significant improvement in symptoms without obvious adverse effects [342,343,344].

## 9. Conclusions

The intracellular transduction of exosomes, including miRNAs and proteins, is regulated by clathrin-dependent or -independent endocytosis and micropinocytosis. By controlling the organ or tissue tropism of exosomes and promoting their intracellular translocation, we expect to construct safe and effective drug delivery systems. Moreover, modification of the peptides on the surface of the exosomes is a novel technique to generate drug-encapsulating exosomes. Furthermore, modified exosomes whose surfaces are modified with binding ligands can be expected to be highly safe and result in effective transduction of DDS by controlling tissue tropism in vivo. Thus, it is expected to become a touchstone for the establishment of new treatment strategies for various diseases, including cancer, especially for intractable diseases for which no treatment methods have been available thus far.

## Figures and Tables

**Figure 1 ijms-23-12971-f001:**
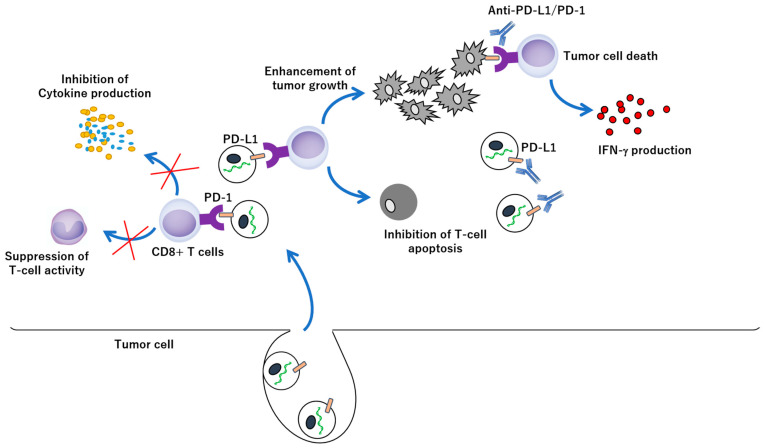
Exosomal PD-L1 in tumour growth and anti-PD-1/PD-L1 therapy. Exosomal PD-L1 secreted by tumour cells is leading to enhancement of tumour growth by reduction of T cell activity and inhibition of cytokine production, including IFN-γ and IL2, and limit effectiveness of anti-PD-1/PD-L1 therapy through binding to antibodies. On the other hand, elimination of the exosomal PD-L1 improves anti-PD-1/PD-L1 therapy.

**Figure 2 ijms-23-12971-f002:**
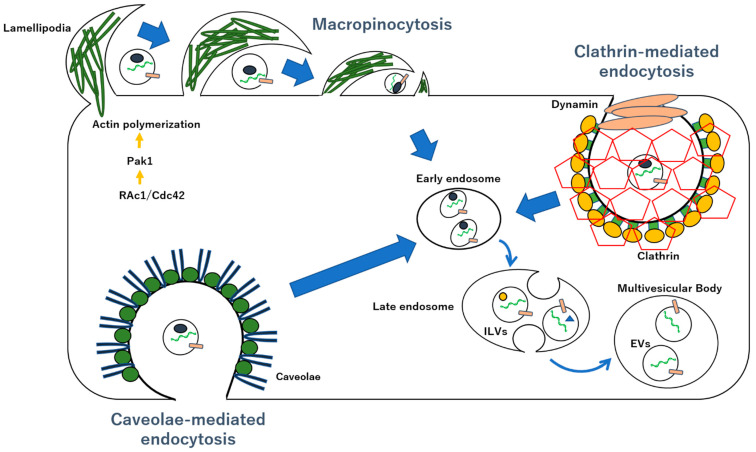
Uptake mechanisms for the transport of EVs. Macropinocytosis is characterized by signal transduction involving the activation of the small G-protein Rac, which leads to polymerization of the actin backbone and formation of lamellipodia in the plasma membrane. Utilizing the lamellipodia structure of the plasma membrane, cells usually surround the extracellular fluid with a size of more than 1 μm, eventually forming vacuoles and taking them into the cell. Clathrin-mediated endocytosis involves five steps, including depression formation of membrane, accumulation of cargo, membrane encapsulation (formation of clathrin-coated pits), cutting, and uncoating, to deliver membrane vesicles containing cargo into the cell. The formation of caveolar endocytic vesicles requires the oligomerization of caveolin, which leads to the formation of caveolin-rich microdomains in the plasma membrane. ILVs: intralumenal vesicles.

## Data Availability

Not applicable.

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
