# Peer review of "Molecular Docking and Intracellular Translocation of Extracellular Vesicles for Efficient Drug Delivery"

_ijms, 2022, doi:10.3390/ijms232112971_

Round 1

Reviewer 2 Report

The authors have submitted an interesting review article entitled "Molecular Docking and Intracellular Translocation of Extracellular Vesicles and their Use as Drug Delivery System" which summarizes the recent works on efficient delivery via leveraging molecular docking and intracellular translocation of exosomes. The manuscript reads well overall, although it will need a spelling and style check. I suggest this article be published after a major revision.

Comments:

 1- First of all, I would like to recommend authors to design a better “Graphical Abstract” for this study to better show the whole story in a simple and informative manner.

2- Moreover, I would like to suggest an intuitive title for this review paper “Leveraging Molecular Docking and Intracellular Translocation of Extracellular Vesicles for efficient drug delivery”

3- Please be sure you have all “permission”s for adopted figures for this paper.

 4- Please carefully revise the manuscript to remove possible grammatical errors and vague sentences. Please thoroughly double-check the whole manuscript.

5- Some of the references in the introduction part are too old (e.g., 2001) and it is not acceptable at all. A myriad of research bodies has been published in recent years and you can find similar concepts and cite them in your paper rather than more than 2 decades old references. Moreover, on page 4, line 158, please read and add valuable information from the following papers as well: 

https://doi.org/10.1016/j.jconrel.2015.07.019, https://doi.org/10.3390/ijms23042223

6- The conclusion section is too short. Please develop it more and add future perspectives to this section as well.  

7- I also suggest the authors devote a certain section to the “clinical application” of this technology and briefly discuss it.

Round 2

Reviewer 2 Report

The manuscript is well-amended and it is ready for publication. I have no further comments.